# Effect of Pyrolysis Temperature on the Characterisation of Dissolved Organic Matter from Pyroligneous Acid

**DOI:** 10.3390/molecules26113416

**Published:** 2021-06-04

**Authors:** Genmao Guo, Qingqing Wang, Qing Huang, Qionglin Fu, Yin Liu, Junfeng Wang, Shan Hu, Ondřej Mašek, Luya Wang, Ju Zhang

**Affiliations:** 1College of Ecology and Environment, Hainan University, Haikou 570228, China; guogenmao123@163.com (G.G.); wangqingqing199301@163.com (Q.W.); fuqionglin99@163.com (Q.F.); liuyin050@163.com (Y.L.); drjunfengwang2010@163.com (J.W.); hushan2000@126.com (S.H.); wangluya8755299@163.com (L.W.); 2Key Laboratory of Agro-Forestry Environmental Processes and Ecological Regulation of Hainan Province, Haikou 570228, China; 3Center for Eco-Environmental Restoration Engineering of Hainan Province, Haikou 570228, China; 4State Key Laboratory of Marine Resource Utilization in South China Sea, Hainan University, Haikou 570228, China; 5Key Laboratory for Environmental Toxicology of Haikou, Hainan University, Haikou 570228, China; 6UK Biochar Research Centre, School of Geosciences, University of Edinburgh, Edinburgh EH9 3FF, UK; ondrej.masek@ed.ac.uk; 7Hainan Guowei Eco-Environment Co., Ltd., Haikou 570203, China; zhangjujie@163.com

**Keywords:** pyroligneous acid, dissolved organic matter, two-dimensional correlation spectroscopy, pyrolysis temperature

## Abstract

Dissolved organic matter (DOM) greatly influences the transformation of nutrients and pollutants in the environment. To investigate the effects of pyrolysis temperatures on the composition and evolution of pyroligneous acid (PA)-derived DOM, DOM solutions extracted from a series of PA derived from eucalyptus at five pyrolysis temperature ranges (240–420 °C) were analysed with Fourier transform infrared spectroscopy, gas chromatography–mass spectroscopy, and fluorescence spectroscopy. Results showed that the dissolved organic carbon content sharply increased (*p* < 0.05) with an increase in pyrolysis temperature. Analysis of the dissolved organic matter composition showed that humic-acid-like substances (71.34–100%) dominated and other fluorescent components (i.e., fulvic-acid-like, soluble microbial by-products, and proteinlike substances) disappeared at high temperatures (>370 °C). The results of two-dimensional correlation spectroscopic analysis suggested that with increasing pyrolysis temperatures, the humic-acid-like substances became more sensitive than other fluorescent components. This study provides valuable information on the characteristic evolution of PA-derived DOM.

## 1. Introduction

Pyrolysis is increasingly becoming the most attractive technology for converting biomass waste into bio-oil or biochar [1]. Pyroligneous acid (PA), a by-product of biomass biochar, contains organic substances such as phenolics, aldehydes, ketones, esters, and acids [2,3]. PA has been widely applied as a bacteriostatic agent, plant growth promoter, antioxidant agent, and feed additive because of its complex composition [4,5].

PA and biochar, which are carbon-rich substances with abundant functional groups, have been produced through the pyrolysis of biomass including agricultural and forestry residues in the absence of oxygen [6,7]. Previous studies indicated that biochar-derived dissolved organic matter (DOM) shows significantly different environmental behaviours and recalcitrance because of its abundant reactive functional groups, including phenolic, hydroxyl, and carboxyl groups [8,9]. The properties of biochar-derived DOM, such as aromaticity, humification degree, and functional group types, have been shown to highly depend on the pyrolysis temperature during biochar preparation [10]. Another study revealed that the binding site between the heavy metal and biochar-derived DOM fluorescent substances differs, and that biochar-derived DOM greatly influences the chemical adsorption of hydrophobic organic pollutants [11]. DOM greatly influences the transformation of nutrients and pollutants in the environment. Previous studies have shown that humic substances exhibit the most negative effect on arsenate adsorption compared to polysaccharide and protein [12]. Wu et al. [13] found that the humic substance components were responsible for Cu(II) binding, and the protein was the only fraction involved in Cd(II) binding. Besides, proteobacteria were closely associated with DOM components during the DOM-degrading process [14,15].

Various spectroscopic techniques, including excitation–emission matrix (EEM), ultraviolet–visible, synchronous fluorescence, and nuclear magnetic resonance spectroscopy, have been conducted to identify the structure and components of DOM [16]. Two-dimensional correlation spectroscopy (2D-COS) can provide insights into changes in the DOM molecular structure because of its ability to prevent overlapping peaks by extending the spectra into the second dimension [17]. The major probes used in 2D-COS are Fourier transform infrared (FTIR), synchronous fluorescence, ultraviolet–vis absorption, and fluorescence, each of which give specific molecular information on the sequential orders of structural variations in response to external perturbations, such as pyrolysis temperature, pH, and composting time [18]. However, few studies have focused on the evolutionary components of PA-derived DOM under different pyrolysis temperatures using EEM fluorescence measurements and 2D spectral analysis.

In this study, eucalyptus was collected as feedstock and pyrolysed at different temperatures of 240–420 °C for PA production. The composition and structure of DOM in PA were analysed by EEM spectroscopy, gas chromatography–mass spectroscopy (GC-MS), and FTIR spectroscopy. Moreover, fluorescence regional integration (FRI) and 2D-COS were performed. The specific aims of this study were as follows: (1) to determine the strength and sequence of the functional group changes in PA-derived DOM at different temperatures; (2) to evaluate the components during the evolution of DOM at different pyrolysis temperatures; (3) to investigate the effects of pyrolysis temperature on PA compositions.

## 2. Results and Discussion

### 2.1. Difference in DOM of PA at Different Temperature Ranges Characterised via Excitation–Emission Matrix-Fluorescence Regional Integration

#### 2.1.1. Fluorescent Components

The fluorescence EEM spectra of PA-derived DOM at different temperature ranges are shown in Figure 1. According to Chen et al. [19] and Song et al. [20], EEM spectra can be divided into five E_x_/E_m_ regions, such that the fluorescence peaks assigned in regions I (E_x_/E_m_: 200–250/280–330 nm) and II (E_x_/E_m_: 200–250/330–380 nm) are related to proteinlike matters, such as tryptophan and tyrosine. Fluorescence peaks related with soluble microbial by-products and fulvic-acid-like and humic-acid-like substances are in regions IV (E_x_/E_m_: 250–450/280–380 nm), III (E_x_/E_m_: 200–250/380–550 nm), and V (E_x_/E_m_: 250–450/380–550 nm), respectively.

As the pyrolysis temperature was increased, the absorbance of the fluorescence peak first increased and then decreased. In the PA-derived DOM fluorescence spectra, dominant strong peaks of humic-acid-like substances and soluble microbial by-products were detected in regions IV and V, respectively, and the peak intensities of regions I, II, and III were relatively weak (Figure 1A–E). To quantitatively assess the changes in the distribution characteristics of the five E_x_/E_m_ regions presented in the EEM fluorescence spectra via FRI analysis (Figure 1F), the P_V, n_ values of humic-acid-like substances and soluble microbial by-products were evaluated. High percentages of humic-acid-like substances occurred in region V (P_V,n_: 71.33–100%). Specifically, the P_V,n_ value of humic-acid-like substances was the highest in temperature ranges T4 and T5, followed by the soluble microbial by-products in region IV, with P_IV,n_ values ranging from 15.70% to 27.55%. High pyrolysis temperatures can promote the formation of humiclike substances with aromatic structures [21]. The relative proportion of humiclike components increased with increasing temperature, and the presence of humiclike substances may be related to the degradation of lignin and biomass materials [22]. PA is a by-product of gas condensation during pyrolysis in biochar production, involving complicated chemical components [23,24]. Moreover, PA contains phenolic substances with aromatic structures [25]. The P_III_,_n_ values of fulvic-acid-like substances decreased from 5% to 0.03% when the pyrolysis temperature ranges were increased from T1 to T3. As the pyrolysis temperatures increased, the P_III_,_n_ values of fulvic-acid-like substances decreased to 0%. These results indicate that fulvic-acid-like substances in PA-derived DOM were present at a low temperature range and the peaks of fulvic-acid-like substances disappeared at a high temperature range, which is consistent with the results of Uchimiya et al. [26], who demonstrated that fulvic-acid-like substances nearly disappear at high pyrolysis temperatures. As the pyrolysis temperature increases, fulvic-acid-like substances easily decompose. In summary, the pyrolysis temperature significantly changed the distribution of the different components of DOM derived from PA; fulvic-acid-like substances were generated at low pyrolysis temperatures (240–370 °C) and their contents gradually decreased with increasing temperature, whereas at high temperatures (370–420 °C), DOM was predominately composed of humic-acid-like substances. 

#### 2.1.2. Variations in DOM Quality Indices

We observed differences in PA-derived DOM quality indices (Fn (335), HIX, FI, and BIX) at different pyrolysis temperature ranges (Table 1). The mean HIX values of the PA-derived DOM ranged between 0.06 and 8.2. HIX values distinguished the DOM of different humification degrees and sources [27,28,29,30]. Speratti et al. [31] indicated that high HIX ratios explain the occurrence of highly humified organic substances based on the presence of complex aromatic molecules, confirming our finding that the humification degree of DOM was greater at lower (240–340 °C) than at higher (340–420 °C) temperatures. The HIX value of PA-derived DOM first increased and then sharply decreased with increasing pyrolysis temperature. FI values of ~1.4 correspond to DOM mainly derived from terrestrial sources, such as plants and soil organic matter, whereas a value of ~1.9 corresponds to DOM from microbial sources in soil [32,33]. The mean FI values ranged from 1.66 to 2.44, indicating increased microbial activity in PA-derived DOM. However, the effect of microbial activity on the formation of PA-derived DOM was weak during biomass pyrolysis. The mean Fn (335) values ranged from 279.45 to 4219.92, and the mean BIX values (except T1 and T2) were all <1, indicating that the autochthonous characteristics of the PA-derived DOM were not obvious, which showed that variations in the HIX, FI, and BIX indices are influenced by the temperatures during biomass pyrolysis.

#### 2.1.3. Variations in DOC Released from PA at Different Temperature Ranges

The content of DOM solution can be expressed by DOC content [34]. The DOC contents released from PA produced at different pyrolysis temperature ranges, as shown in Table 1, varied significantly from 699.3 ± 0.91 to 1532.66 ± 11.59 mg L^−1^. The DOC content of the temperature range T5 (1532.66 ± 11.59 mg L^−1^) was the highest. These observations are consistent with those of a previous study showing that the content of DOC increases with increasing pyrolysis temperatures, which may be because with an increasing pyrolysis temperature, the volatile matter content in PA gradually increases, leading to elevations in the corresponding DOC content. Phenolics, which were the primary high-molecular-weight components, were the primary components in PA, possibly because lignin broke at high pyrolysis temperatures, resulting in greater production of phenolics [35,36]. The abundant formation of phenolics at high pyrolysis temperatures may explain the reason DOC content increased as the pyrolysis temperature increased; the DOC content was the highest at high pyrolysis temperatures in this study. Additionally, the feedstock may have also influenced DOC content during biomass pyrolysis. Uchimiya et al. [26] found that the sequence of changes in the DOC content in different feedstock was shell > pecan shell > broiler litter > cottonseed hull under the pyrolyzed temperature (500 °C). Jamieson et al. [22] reported that the difference in DOC content of yellow birch biochar (353 BS) pyrolysed at 353 °C was significantly higher than that of sugar maple biochar (380 MS) pyrolysed at 353 °C (*p* < 0.05). 

### 2.2. Two-Dimensional Correlation Spectroscopic Analysis

#### 2.2.1. Two-Dimensional Correlation Analysis of FTIR

Changes in the chemical molecular structure of PA during pyrolysis were characterised by FTIR spectroscopy. The absorption peak at 3600–3300 cm^−1^ is associated with a phenolic hydroxyl group, and its intensity decreased significantly with increasing temperature. The absorption peak at 2970–2860 cm^−1^ corresponds to the -CH_2_ alkane symmetrical stretching. The absorption peak at 1750–1600 cm^−1^ corresponds to a C=O stretching vibration with carboxyl groups, aldehydes, and ketones, and an intense C-O stretch was observed at around 1200–1000 cm^−1^. 

There were overlapping peaks and poor identification of molecular structures in key spectral regions in the traditional FTIR spectra. To study the molecular structure of PA components in detail, the FTIR spectra were divided into three ranges: 900–700, 1800–900, and 3600–2800 cm^−1^. The 2D-COS synchronous (a, c, and e) and asynchronous (b, d, and f) spectra are shown in Figure 2. In the 3600–2800 cm^−1^ range, autopeaks in the synchronous spectra (Φ (ν_1_, ν_2_)) at 3442 and 2983 cm^−1^, corresponding to free O-H and alkane -CH_2_ signals, respectively, showed that these groups were most sensitive to temperature change (Figure 2a). In addition, there was a cross-peak Ψ (3055, 3392) in the asynchronous Ψ (ν_1_, ν_2_) spectrum (Figure 2b). According to Noda’s rules, the positive cross-peak at Φ (2988, 3461) indicated that the C-H and H-O bonds of aliphatic chains exhibit the same trends, which were mainly attributed to the dehydration of pyran rings and ring fragments and indicates that the bond breaking of hydroxyl and methylene reactions of the side chains of the benzene rings are synchronous [37]. Similar results revealed that the cross-peak at Φ (3425, 2925) between -CH_2_ (2925 cm^−1^) and O-H (3425 cm^−1^) corresponds to synchronous reactions that occur during the pyrolysis of hemicellulose and compounds containing -CH_2_OH groups [38].

For the band at 1800–900 cm^−1^ in the synchronous map, the main autopeaks observed at 984, 1185, 1295, 1601, and 1684 cm^−1^ (Figure 2c) were positive, indicating a continuous change in PA functional groups with changes in the pyrolysis temperature. These six cross-peaks were also found in the diagonal on the asynchronous map (Figure 2d). The sign of each cross-peak on the synchronous and asynchronous maps of PA are shown in Table 2, and the sequence of evolution of the functional groups was 1601 > 1185 > 1295 > 1684 > 984 cm^−1^, corresponding to C=O stretching of ketone → stretching vibrations of aliphatic C-O → C-OH stretching of the carboxylic acid group → C=C stretching of aromatic ring → C-H stretching of aromatic structures. The peaks at 1601 and 1185 cm^−1^, corresponding to the C=O stretching of quinone, or ketone, and C-O stretching of polysaccharides, respectively [39], changed at an earlier time point. This suggests that proteinlike substances were significantly decomposed and ketones were generated [40]. The peaks at 1295 and 1684 cm^−1^ were attributed to C-OH stretching of the carboxylic acid group and C=C stretching of the aromatic rings, respectively [41]. C-H stretching of the aromatic structures indicates high aromaticity, owing to which they are not easily decomposed [42]. A similar study revealed structural changes in the DOM as the pyrolysis temperature increased; the order of peak changes was polysaccharide C-O > lipid group C=O > carboxyl group C=O > aromatic ring C-H > aromatic ring C=C [43]. 

In the 900–700 cm^−1^ band, autopeaks in the synchronous spectra Φ (832, 832), Φ (775, 775), and Φ (880, 880) corresponded to the out-of-plane bending vibration of benzene C-H, indicating that the trends were the same as those observed with increasing temperature during pyrolysis. In the synchronous spectrum Φ (ν_1_, ν_2_), the cross-peaks Φ (880, 832), Φ (880, 750), Φ (832, 775), and Φ (832, 750) were all positive, whereas Ψ (880, 832) was negative in the asynchronous spectrum Ψ (ν_1_, ν_2_). According to Noda’s rules, 1,4-substituted benzene (832 cm^−1^) was formed before 1,2,3(4),5-substituted benzene (880 cm^−1^) (Figure 2e,f).

#### 2.2.2. Two-Dimensional Correlation Fluorescence Spectroscopy

2D-COS analysis helps to enhance the deconvolution of overlapping peaks and provides information on the heterogeneous distribution of PA-derived DOM with the pyrolysis temperature as an external perturbation [44]. The synchronous fluorescence spectra of PA in the 200–600 nm region (Figure 3A) exhibited three predominant autopeaks at 450, 420, 380, and 360 nm with a small peak at 320 nm identified from the cross-peaks. All of these cross-peaks were positive, indicating that the spectral changes proceeded in the same direction as the temperature variation. The asynchronous map provided information on the sequential relationship between two spectra. As shown in Figure 3A and Table 3, the cross-peaks located in the corner of the asynchronous Ψ (ν_1_, ν_2_) spectrum showed negative signs except for Ψ (360, 380), demonstrating that according to Noda’s rule, PA-derived DOM fractions changed in the following order with changes in temperature: 450 > 360 nm and 380 > 320 nm. This corresponded to the following sequence: humiclike fraction → fulviclike fraction → proteinlike fraction. The proteinlike peak was in the 250–330 nm band, which was attributed to the presence of proteinaceous and aromatic substances. The humic-acid-like peak was in the 380–500 nm and fulvic-acid-like in the 330–380 nm region [45]. Proteinlike substances were representative of the biodegradable and humiclike substances and can be referred to as non-biodegradable components [46]. Similar studies found that the order of changes in the three components’ different feedstocks with increasing pyrolysis temperatures was proteinlike=humic acid → fulvic acid for chicken biochar, and fulvic acid → humic acid → proteinlike for dairy manure biochar [44]. Baken et al. [47] indicated that with changes in the pyrolysis temperature, humiclike substances responded faster than proteinlike substances because of the higher aromaticity of the former. The fulviclike fraction was more susceptible than the other fractions (e.g., proteinlike and humiclike fractions). The amount of proteinlike substances decreased as that of the humic-acid-like substances increased with increasing pyrolysis temperatures in this study, and humic-acid-like substances were the main fluorescence components in PA. Wu et al. [48] indicated that the decomposition of proteinlike substances is important for humification. Carboxyl acids are the main precursors of humiclike substances. The asynchronous map showed that the humiclike fraction was produced earlier than the other fluorescence components.

### 2.3. PA Components at Different Temperature Ranges

A total of 41 chemical compounds was identified and classified, representing 71.19–89.36% of the PA composition and included acids, phenolics, ketones, aldehydes, esters, and others (Figure 4a,b). The relative content of acids decreased from 45.23% to 7.48% as the pyrolysis temperature increased from T1 to T5. The relative content of phenolics ranged from 51.04–62.35% and that of ketones, aldehydes, esters, and others from 10.7–21.01% of the total PA composition. Specifically, the major chemical components of PA were acetic acid (2.75–10.23%), 3-methoxy-1,2-benzenediol (11.51–14.59%), 2,6-dimethoxyphenol (15.02–20.51%), catechol (8.71–15%), and 3,5-dimethoxy-4-hydroxytoluene (2.55–9.96%) when the pyrolysis temperature ranges were increased from T1 to T5, respectively. The 2,6-dimethoxyphenol content was the highest; this component is generated from the vigorous decomposition of lignin with various hydroxyl- and methoxy-substituted structures during pyrolysis, acetic acid is generated from the breakdown of acetyl groups attached to xylan units from hemicellulose dehydration [49]. Taken together, these results show that the pyrolysis temperature greatly influenced PA components.

### 2.4. Relationship between Chemical Components and Spectral Parameters

Correlation heat maps were used to describe the relationship between DOC, fluorescence indices (HIX, BIX, FI, Fn (335)), fluorescence components (I–V), FTIR spectral parameters (1504/1601, 1504/1684, 1504/1295, 1504/1185), and the chemical components of PA at different temperatures (Figure 5). HIX was significantly positively correlated with aldehydes (*p* < 0.01), esters (*p* < 0.01), and phenolics (*p* < 0.05) and negatively correlated with alcohols (*p* < 0.05). The fluorescent components I, II, and III exhibited a significant negative correlation with acids (*p* < 0.05). The fulvic-acid-like fraction and humic-acid-like fraction were related to esters, carboxylic, and phenolic groups [50]. Phenolics in the fulvic-acid-like fraction can interact with amino acids to produce humus substances, further explaining the DOM humification degrees [29]. The degradation products of protein substances are the dominant precursors for humification [48].

## 3. Materials and Methods

### 3.1. Pyroligneous Acid Production

Raw PA was prepared from eucalyptus using a traditional black charcoal kiln and collected by running water through a shuttle connected to the chimneys to condense the smoke. PA was collected over a temperature range of 240–420 °C, including five more specific temperature ranges: T1, 240–270 °C; T2, 270–340 °C; T3, 340–370 °C; T4, 370–400 °C; T5, 400–420 °C. All PA samples were stored in sealed glass bottles for further analysis.

### 3.2. Excitation–Emission Matrix Fluorescence Measurements and Fluorescence Regional Integration Analysis

Before determining the dissolved organic carbon (DOC) content, all PA samples were diluted to ~10 mg mL^−1^ to minimise inner filtering, and then analysed using a total organic carbon analyser [51]. The fluorescence EEM spectra of all diluted samples were determined using a HIACHI F-4600 fluorescence spectrometer across excitation (E_x_) wavelengths of 200–450 nm (every 5 nm) and emission (E_m_) wavelengths of 200–600 nm (every 5 nm) at room temperature. The EEM spectra were recorded at a scan rate of 2400 nm min^−1^ and the slit widths of both the E_m_ and E_x_ wavelengths were set to 5 nm. For the blank scans, Milli-Q water was used at intervals of every 10 runs. The 3D fluorescence spectral data of the sample were subtracted from those of the blank ultrapure water before analysis, thus eliminating the influence of Rayleigh and Raman scattering. EEM spectral data were analysed using the FRI method, which delineated EEM into five E_x_-E_m_ regions. Fluorescence intensity was integrated beneath each of the five EEM regions, and the percent fluorescence response (P_i,n_) was calculated for reference.
(1)$$\varphi_{I}=\int_{\mathrm{ex}}\int_{\mathrm{em}}I\left(\lambda_{\mathrm{ex}}\lambda_{\mathrm{em}}\right){d\lambda }_{\mathrm{ex}}{d\lambda }_{\mathrm{em}},$$
(2)$$\Phi_{i,n}={\mathrm{MF}}_{i}\times \Phi_{i},$$
(3)$$P_{i,n}=\Phi_{i,n}/\Phi_{T,n}\times 100\%,$$
where Φ_I_ is the integral volume of fluorescence region i (au·), λ_ex_ is the E_x_ wavelength (nm), λ_em_ is the E_m_ wavelength (nm), and I(λ_ex_λ_em_) is the fluorescence intensity at each E_x_/E_m_ wavelength pair; Φ_i,n_ is the integral standard volume of the fluorescent area i (au), MF_i_ is the multiplication factor, and Φ_T,n_ is the integral standard volume of the total fluorescent area (au); and P_i,n_ is the ratio (%) of the integrated standard volume of a certain fluorescence area (i) to the total integrated standard volume, respectively. Spectral parameters, including the fluorescence index (FI), autochthonous index (BIX), Fn (355), and humification index (HIX), were calculated (Table 4) [52,53].

### 3.3. Fourier-Transform Infrared Spectroscopy

FTIR spectra of the PA samples were recorded on a FTIR-650 spectrometer (Gangdong, Tianjing, China) at wavelengths of 4000–400 cm^−1^, with the samples coated in KRS-5 crystals. The resolution was 4 cm^−1^ and each sample was scanned 32 times to obtain an average spectrum. The ratio between the peak intensity of the chemical functional groups (I_1504/1601_, I_1501/1684_, I_1501/1295_, and I_1501/1185_) was calculated from the FTIR spectra.

### 3.4. Gas Chromatography–Mass Spectroscopy

Organic components of PA were analysed by GC-MS with an HP-5MS (30 m × 0.25 mm × 0.25 μm). Helium was used as a carrier gas at a constant flow rate of 1 mL min^−1^. The GC-MS conditions were as follows: initial temperature of 60 °C, increased to 180 °C at a rate of 15 °C min^−1^ and held for 1 min, and then increased to 270 °C at a rate of 10 °C min^−1^ and held for 5 min, split injection at a split rate of 10:1 with a delayed time of 2.5 min, ion source temperature of 230 °C, and mass scanning range of 20–550 amu s^−1^. Chemical compounds were identified by comparison of the spectra with those from the National Institute of Standards and Technology database.

### 3.5. Two-Dimensional Correlation Analysis

For further analysis, 2D-COS analysis was employed using excitation–emission fluorescence and FTIR spectra, with the pyrolysis temperature as the external perturbation from the PA sample. Thereafter, a 2D correlation spectrum matrix was obtained using MATLAB.22 and Origin 8.5 software to mathematically transform the data into processed infrared spectra [50]. Better selectivity is achieved when changes in the spectral information are identified based on specific external disturbances, composed of synchronous and asynchronous spectra. Functional groups and fluorescence component were identified according to the positive/negative and presence/absence of cross-peaks in the asynchronous and synchronous spectra (Table 5) [54,55]. 

## 4. Conclusions

This study revealed the different characteristics of PA-derived DOM at different pyrolysis temperatures through chemometric EEM–FRI analysis and 2D-COS. Humic-acid-like material was the key PA-derived DOM component, and the DOC content increased with increasing pyrolysis temperatures. Moreover, a higher humification degree of DOM was observed at lower temperatures than at higher temperatures. With increasing pyrolysis temperatures, humic-acid-like material in the DOM and chemical functional groups for C=O stretching showed the earliest changes. This study provides a valid alternative for assessing the quantity and quality of PA-derived DOM and potential environmental applications.

## Figures and Tables

**Figure 1 molecules-26-03416-f001:**
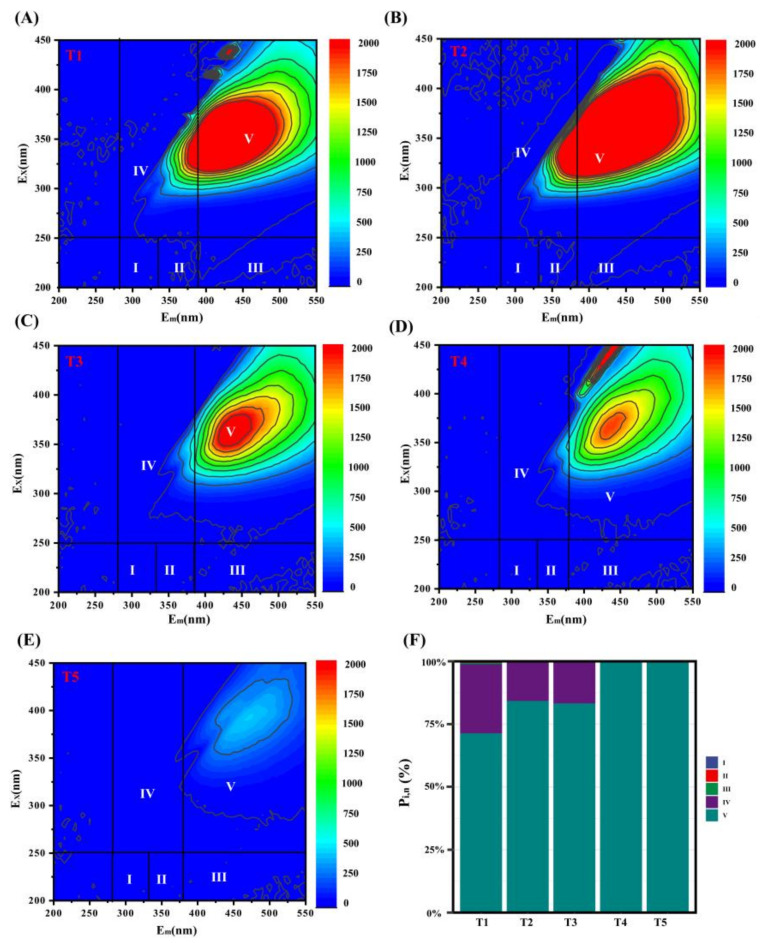
Fluorescence excitation–emission matrix spectra (**A**–**F**) and distribution of FRI of PA-derived DOM samples at different temperature ranges (T1, 240–270 °C; T2, 270–340 °C; T3, 340–370 °C; T4, 370–400 °C; T5, 400–420 °C).

**Figure 2 molecules-26-03416-f002:**
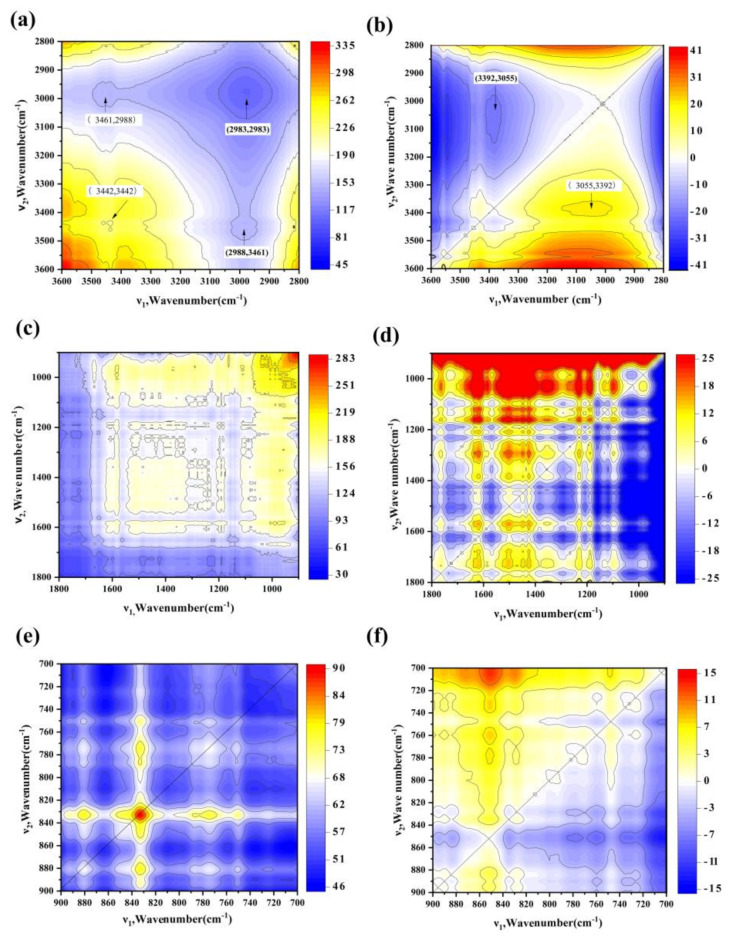
2D-COS analysis at wavelengths of 3600–2800, 1800–900, and 900–700 cm^−1^ for the PA synchronous (**a**,**c**,**e**) and asynchronous (**b**,**d**,**f**) spectra. Red and blue areas represent positive and negative correlation values, respectively, whereas the white area represents zero correlation values.

**Figure 3 molecules-26-03416-f003:**
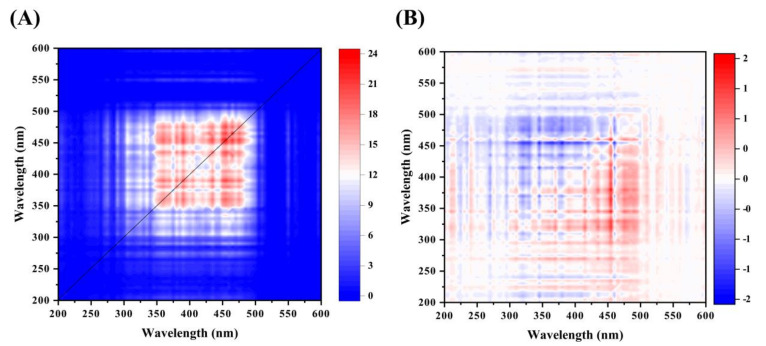
Synchronous (**A**) and asynchronous (**B**) 2D correlation maps generated from the synchronous fluorescence spectra of PA-derived DOM.

**Figure 4 molecules-26-03416-f004:**
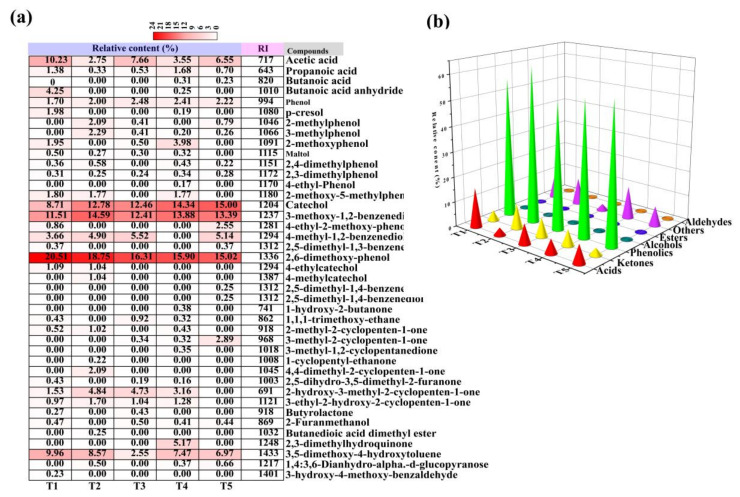
Effects of the pyrolysis temperature on the organic components of pyroligneous acid (PA). (**a**) Chemical compounds. (**b**) Organic component types.

**Figure 5 molecules-26-03416-f005:**
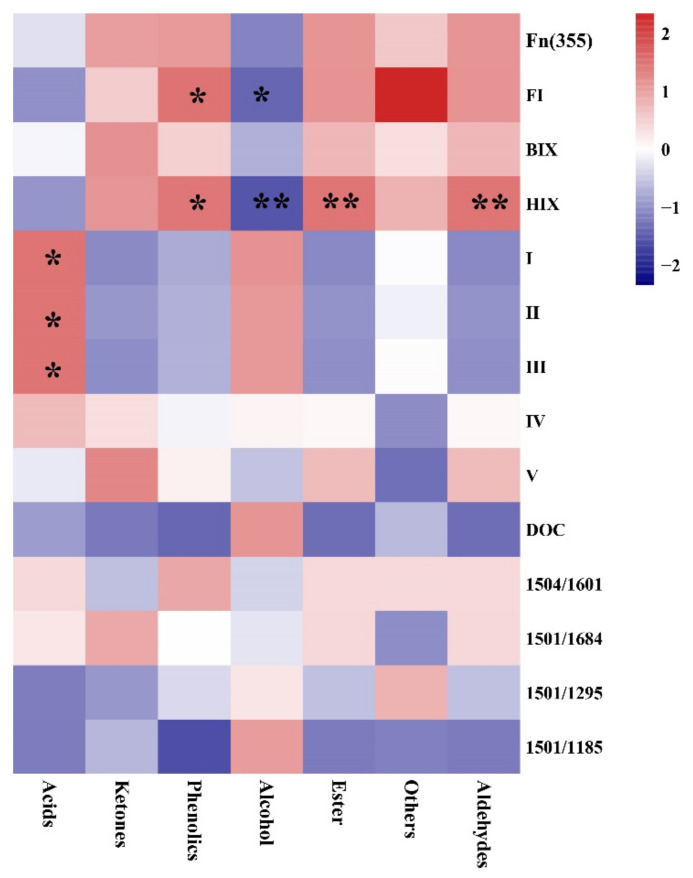
Relationship between DOC, fluorescence spectral indices (HIX, BIX, FI, and Fn (335)), fluorescence components (I–V), FTIR spectral parameters (1504/1601, 1504/1684, 1504/1295, 1504/1185), and chemical components. * Correlation was significant at the 0.05 level; ** Correlation was significant at the 0.01 level.

**Table 1 molecules-26-03416-t001:** Indices (Fn (335), HIX, FI, and BIX) and dissolved organic content of dissolved organic matter derived from pyroligneous acid at different temperature ranges.

Temperature	DOC (mg/L)	Fn (335)	FI	BIX	HIX
T1	699.30 ± 0.91 b	4219.92 ± 131.08 b	1.91 ± 0.01 b	1.13 ± 0.00 b	1.66 ± 0.02 b
T2	855.46 ± 4.28 a	6549.58 ± 252.42 a	2.24 ± 0.04 a	1.38 ± 0.03 a	8.20 ± 4.81 a
T3	1030.33 ± 4.04 c	1931.14 ± 95.88 c	1.66 ± 0.02 c	0.81 ± 0.03 c	0.06 ± 0.03 c
T4	1245.33 ± 17.00 d	1203.14 ± 785.21 d	1.96 ± 0.04 b	0.64 ± 0.55 c	0.16 ± 0.18b c
T5	1532.66 ± 11.59 e	279.45 ± 34.52 e	1.90 ± 0.04 b	0.08 ± 0.13 d	0.40 ± 0.02b c

Values with different letters significantly differed in rank (*p* < 0.05).

**Table 2 molecules-26-03416-t002:** 2D-FTIR-COS results on the assignment and sign of each cross-peak in synchronous and asynchronous maps of PA samples at different temperatures (1800–900 cm^−1^).

Position (cm^−1^)	984	1185	1295	1601	1684
984	+	+(−)	+(−)	+(−)	+(−)
1185		+	+(+)	+(−)	+(+)
1295			+	+(+)	+(+)
1601				+	+(+)
1684					+

**Table 3 molecules-26-03416-t003:** 2D-COS fluorescence analysis results of the sign of each cross-peak in the synchronous (Φ) and asynchronous (Ψ) maps of PA-derived DOM samples at different temperatures.

Position (cm^−1^)	320	360	380	450
320	+	+(−)	+(−)	+(−)
360		+	+(+)	+(−)
380			+	+(−)
450				+

**Table 4 molecules-26-03416-t004:** Description of fluorescence spectrum parameters.

Index	Definition
Fn (355)	Fluorescence signal intensity at E_x_=355 nm, E_m_=450 nm
HIX (Humification index)	Region integral ratio between E_m_=435–480 nm and E_m_=300–345 nm at E_x_=245 nm.
FI (Fluorescence index)	E_x_=370 nm, ratio of between E_m_=470 nm and E_m_=520 nm.
BIX (Autochthonous index)	Ratio of fluorescence intensity at E_m_=380–430 nm at E_x_=310 nm

**Table 5 molecules-26-03416-t005:** Noda’s rule and method for determining crossing peaks in 2D-COS.

Ψ (ν_1_, ν_2_)	Φ (ν_1_, ν_2_)	Interpretation
	+	Intensity of ν_1_ and ν_2_ are changing in the same direction
	−	Intensity of ν_1_ and ν_2_ are changing in the opposite direction
+	+	Change at ν_1_ is occurring predominantly before that at ν_2_
−	+	Change at ν_1_ is occurring predominantly after that at ν_2_
−	−	Change at ν_1_ is occurring predominantly before that at ν_2_
+	−	Change at ν_1_ is occurring predominantly after that at ν_2_

## Data Availability

Not applicable.

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
