# Peer review of "Effect of Pyrolysis Temperature on the Characterisation of Dissolved Organic Matter from Pyroligneous Acid"

_molecules, 2021, doi:10.3390/molecules26113416_

Round 1
Reviewer 1 Report
It is a paper that analyzes organic substances that are dissolved in pyroligneous acid according to pyrolysis temperature during biochar production process.
Using analyzers, the distribution of three main substances, humic acid like substances, fulvic acid like substances, microbial products was revealed.
It is believed to provide useful information to readers.
A few simple amendments are needed.
- Page 5, Session 3.1.2, 2nd line; I think it's BIX, not HIX. The numbers presented are BIX related, so please review them.
- What does a-e mean in Table 3?
- Session 3.1.3 last line: what does (353BS) and 380MS mean?
- Check typo and session number (session 2.1.1?)
Author Response
Response to Reviewer 1 Comments:
Comments 1: It is a paper that analyzes organic substances that are dissolved in pyroligneous acid according to pyrolysis temperature during biochar production process. Using analyzers, the distribution of three main substances, humic acid like substances, fulvic acid like substances, microbial products was revealed. It is believed to provide useful information to readers.
Response 1: Thank you for your appreciation and recognition.
A few simple amendments are needed.
Comment 2: Page 5, Session 3.1.2, 2nd line; I think it's BIX, not HIX. The numbers presented are BIX related, so please review them.
Response 2: Thank you for your constructive comments and your careful work,which has been revised in Line 185.(The mean HIX values of the PA-derived DOM ranged between 0.06 and 8.2).
Comment 3: What does a-e mean in Table 3?
Response 3: Thank you for your instructive question. We has added the annotation information in the Table 3 (Line 204). Value with the different letters(a-e) are significantly different in a rank (p<0.05).
Comment 4: Session 3.1.3 last line: what does (353BS) and 380MS mean?
Response 4: Thank you for your instructive question. We has modified the information in the Line 225-227. 353BS represented Birch biochar pyrolysed at temperature of 353℃. 380MS represented sugar maple biochar was pyrolysed at temperature of 380℃.
Comment 5: Check typo and session number (session 2.1.1?)
Response 5: Thank you for your kindness. This manuscript has been revised and polished by professionally specialist of the native English (session 2.1.1).
In a word, thank you for your hard work on our manuscript.

Reviewer 2 Report
This article entitled "Effect of Pyrolysis Temperature on Characterisation of Dissolved Organic Matter from Pyroligneous Acid" has been carefully reviewed. However, many typos and grammatical errors make the manuscript difficult to understand.
In abstract, humic acid-like or humic-acid like; fourier should be Fourier; what is "protein-like" ?
Conflflicts, typo;
Too much repetitive information bothers the reader. For example, the legend in the right side of figure 1(F) seems to show that part of the data is hidden. And, figure 4 is completely incomprehensible.
It is recommended that the authors reorganize the paper and briefly present the experimental results more clear.
Author Response
Response to Reviewer 2 Comments:
Comment 1: This article entitled "Effect of Pyrolysis Temperature on Characterisation of Dissolved Organic Matter from Pyroligneous Acid" has been carefully reviewed. However, many typos and grammatical errors make the manuscript difficult to understand.
Response 1: Thank you for your costing time and kindness. This manuscript has been revised and polished by professionally specialist of the native English.
Comment 2: In abstract, humic acid-like or humic-acid like; fourier should be Fourier; what is "protein-like" ?
Response 2: Thank you for your instructive question. Humic-acid like is a macro-molecular organic acids, which composed of aromatics and various functional groups, have good physiological activities and functions such as absorption, complexation, and exchange.
Protein-like is a similar amino acid substance. Meanwhile, this manuscript has been revised and polished by professionally specialist of the native English.
Comment 3: Conflflicts, typo; Too much repetitive information bothers the reader. For example, the legend in the right side of figure 1(F) seems to show that part of the data is hidden. And, figure 4 is completely incomprehensible. It is recommended that the authors reorganize the paper and briefly present the experimental results more clear.
Response 3: Thank you for your kindness comments, we have modified carefully with your comments, and our whole paper has been revised and polished by specialist of the native English.
In a word, thank you for your hard work on our manuscript.

Round 2
Reviewer 2 Report
Thanks to the authors for their efforts to respond to my curiosity and questions about this topic. Pyrolysis plays an important role in the conversion of organic pollutants and nutrients in the environment. This study provides clues about the evolution of the characteristics of PA-derived DOM. It is suggested that the authors can emphasize more the examples of DOM in environmental pollution and nutrient application as an illustration to attract more readers.
Author Response
Response to Reviewer 2 Comments:
Comment 1: Thanks to the authors for their efforts to respond to my curiosity and questions about this topic. Pyrolysis plays an important role in the conversion of organic pollutants and nutrients in the environment. This study provides clues about the evolution of the characteristics of PA-derived DOM. It is suggested that the authors can emphasize more the examples of DOM in environmental pollution and nutrient application as an illustration to attract more readers.m.
Response 1: Thank you for costing your precious time and effort to review our papers, and intersting our study topic. We added more the examples of DOM in environmental pollution and nutrient application as an illustration. The revision in Introduction section as followed:
DOM greatly influences the transformation of nutrients and pollutants in the environment. The previous studies had showed that humic substances exhibited the most negative effect on arsenate adsorption compared to polysaccharide and protein [12, 51, 52]. Wu et al. [53] found that the humic substances components responsible to Cu(II) binding, and the protein was the only fraction involved in Cd(II) binding. Besides, Proteobacteria were closely associated with DOM components during the DOM-degrading process[54, 55].(Line 56-62 and Line 547-558 ).
